
# 30 m Map of Young Forest Age in China

Yuelong Xiao[1], Qunming Wang[1*], Xiaohua Tong[1], Peter M. Atkinson[2,3]

[1]College of Surveying and Geo-Informatics, Tongji University, 1239 Siping Road, Shanghai, 200092, China.
[2]Faculty of Science and Technology, Lancaster University, Lancaster LA1 4YR, UK.
[3]Geography and Environment, University of Southampton, Highfield, Southampton SO17 1BJ, UK.

*Corresponding author*: Qunming Wang(wqm11111@126.com)

**Abstract.** Young forest age mapping at a fine spatial resolution is important for increasing the accuracy of estimating land-atmosphere carbon fluxes and guiding forest management practices. In recent decades, China has actively conducted afforestation and forest protection projects, thereby, laying the foundation for the
realization of carbon neutrality. However, very few studies have been conducted which map the ages of young forests for the whole of China at a fine spatial resolution. In this research, a continuous change detection and classification (CCDC)-based method suitable for large-scale forest age mapping is proposed, and used to estimate young forest ages across China in 2020 at a spatial resolution of 30 m. First, a 10 m spatial resolution land cover dataset (WorldCover2020) from the European Space Agency (ESA) was used to determine the forest
cover areas in 2020. Then, the CCDC algorithm was used to identify stand-replacing disturbances to determine the stand age based on 436,967 Landsat tiles across China from 1990 to 2020. A validation sample set composed of multiple land use/land cover (LULC) products was used to calculate the overall accuracy (OA) of the 2020 young forest age (1–31 years) map of China, and the OA was 90.28%. The reliability and applicability of the proposed CCDC-based forest age mapping method was validated by comparing the forest age map with
Hansen's forest change dataset, Max Planck Institute for Biogeochemistry (MPI-BGC) 1 km global forest age datasets and field measurements. The CCDC-based method has strong application potential in real-time mapping of the age of young forests at the global scale. The produced forest age map provides a basic dataset for research on the forest carbon cycle and forest ecosystem services, and important guidance for government departments, such as the National Forestry and Grassland Administration and National Development and
Reform Commission in China.

## 1 Introduction

The industrial revolution and the use of fossil fuels has led to a continuous increase in the concentration of greenhouse gases, particularly carbon dioxide, in the atmosphere, which has caused an increase in global temperatures. Forest growth plays a significant role in reducing atmospheric carbon dioxide levels, and stand
age has been recognized as an important parameter in forest carbon cycle models (He et al., 2011; Vilen et al., 2012; Zhang et al., 2014). In existing studies, differences in the carbon sequestration capacity of forest stands with different ages have not been considered, which has led to large uncertainties in estimates of carbon sources/sinks in forest ecosystems (Piao et al., 2022). Loboda and Chen (2017) pointed out that young boreal forests (forest age < 30 years) are stable carbon sources, while temperate forests transition from large carbon
sources to significant carbon sinks in the first 10 years until they mature. Therefore, studies on the stand age of young restored forests can contribute to more accurate estimates of forest carbon fluxes.

As a major industrial country, China's carbon dioxide emissions have continued to increase in recent decades, and problems such as land degradation, air pollution and climate change have emerged. To address these problems, China has developed a series of plans to protect and expand its forests (Chen et al., 2019). For
example, in recent decades, China has implemented afforestation and forest conservation projects to restore





natural forests and improve ecosystem services (Lu et al., 2017). Chen et al. (2019) showed that China ranks first in the world in the production of new green areas from 2000 to 2017 and accounted for 25% of the global net increase in leaf area, of which forests contributed the most (42% of China's total greenery). It was found that different land-use changes in southern China increased aboveground carbon stocks by 0.11 ± 0.05 PgC y$^{-1}$

between 2002 and 2017, with 32% of the carbon sink contributed by young forests (Tong et al., 2020). Wang et al. (2020) found that the global contribution of China's forest carbon uptake was underestimated. More precisely, land carbon sinks in southwestern China (Yunnan, Guizhou, and Guangxi provinces) were underestimated throughout the year and land carbon sinks in northeastern China (especially in Heilongjiang and Jilin provinces) were underestimated in the summer months.

Although a large amount of literature has focused on forest cover and carbon sinks in China, few studies have investigated forest age and the spatial distribution of young forests in China. In particular, fine spatial resolution data on forest age are missing. Presently, forest age products in China are available mainly at 1 km spatial resolution. For example, forest age maps of forests and plantations at 1 km spatial resolution in China have been successively produced by Zhang et al. (2014), Zhang et al. (2017) and Yu et al. (2020). However, most forests in

China are distributed in mountainous areas with strong spatial heterogeneity. Generally, the existing forest age data are of too coarse a spatial resolution to support stand calculations for these regions.

The traditional method of forest age mapping is based mainly on field investigation, which is time-consuming and labor-intensive (i.e., it requires considerable human resource and material resources) (Racine et al., 2014), especially in steep mountain forests and areas with inconvenient access. This form of forest age surveying makes

it very difficult to map large areas. In addition, there exist further problems such as poor timeliness and slow updating, which seriously affect the reliability of the collected forest age data (Pan et al., 2011).

Remote sensing images represent a systematic tool for estimating large-scale biophysical variables owing to their wide spatial coverage and frequent data updates (Diao et al., 2020). Generally, the basic physical mechanism for estimating forest age using remote sensing images is that forests of different ages exhibit

different physical characteristics, such as spectral reflectance, tree crown texture, light transmittance and biomass (Champion et al., 2014; Kuusinen et al., 2014; Thom and Keeton, 2019). In particular, regional forest age can be estimated by combining remote sensing data with field survey (such as forest inventory data). The main principle underlying such approaches is that forest age is correlated with the (i) spectral reflectance and/or vegetation index of optical remote sensing images, and (ii) backscattering coefficient and interference

coherence of radar images (Diao et al., 2020). For example, Besnard et al. (2021) used forest inventories, biomass, and climate data to map global forest age around 2010. He et al. (2011) used forest inventory and analysis data to find a threshold for the normalized difference disturbance index to distinguish disturbances from regenerating forests. Combining SPOT 4 satellite sensor data, historical fire data and forest inventory data, Pan et al. (2011) generated a 1 km spatial resolution stand age map for the North American continent. Vilen et

al. (2012) used remote-sensing-based European forest cover data and forest inventory maps to estimate the age of European forests between 1950 and 2010. The relationship between forest age and forest structure (such as tree height) in measured data has also been used to estimate forest age (Racine et al., 2014).

In addition to optical images, Synthetic Aperture Radar (SAR) images play an important role in forest age mapping because of their advantages of all-weather, all-day monitoring. Pinto et al. (2013) found that the

interferometric coherence of the L-band airborne sensor Uninhabited Aerial SAR (UAVSAR) was able to estimate forest age with great accuracy, overcoming the "saturation" problem that occurs in optical image-based forest age mapping. LiDAR data have also been used for forest age mapping. For example, Racine et al. (2014) used airborne LiDAR data and ground data to estimate forest age in Quebec, eastern Canada.

In studies of Chinese forests, age has been widely estimated using the direct relationship between forest age

and tree height. For example, Zhang et al. (2014) constructed the relationship between age and height retrieved from field observations to generate a 1 km spatial resolution map of forest age in China. Zhang et al. (2017) used climate data and forest height data, together with provincial statistical data from the national forest



inventory to produce a downscaling-based 1 km spatial resolution map of forest age distribution in China. Yu et al. (2020) used data such as field measurements, national forest inventory data and remote-sensing-based

forest height maps to map the ages and types of planted forests in China at a spatial resolution of 1 km.

Although the strategy of combining remote sensing data and field survey data has dominated forest age mapping, it still suffers from the following problems. First, the availability of field survey data is difficult to guarantee. The usability problem depends mainly on the positioning accuracy of the sample points, regional differences and the number of samples. The positioning accuracy is affected mainly by measurement errors,

while regional differences are reflected mainly in the differences in data availability caused by various regional policies, laws and regulations. Second, the influence of the saturation phenomenon of spectral reflectance and/or the backscattering coefficient cannot be ignored. This saturation phenomenon means that at large values of forest variables, such as biomass and age, the spectral reflectance and/or backscattering coefficients of remote sensing images are no longer sensitive to changes in these variables (Zhao et al., 2016). For example,

mature forests have a more stable canopy texture and canopy area than young forests. In addition, the saturation problem varies based on stand species and forest structure (Zhao et al., 2016; Lu et al., 2016), which further increases the difficulty in estimating forest age directly from spectral reflectance or backscattering coefficients. Although studies have shown that LiDAR data can solve the saturation problem (Lu et al., 2016), the limited spatial coverage and availability of the observed data hinder widespread application. Third, complex

stand compositions and forest structures make it difficult for a single classification model to achieve reliable forest age mapping. Specifically, the accuracy varies greatly with spectral reflectance, backscattering coefficient, canopy texture and other characteristics of mixed forests.

Methods of estimating forest age based on forest disturbance time can overcome the above problems effectively. This type of method uses time-series images (Powell et al., 2010; Zhu and Liu, 2015; Zhao et al., 2016)

and/or disturbance historical data (such as burn scar maps) to infer the time of the last stand-replacing disturbance to estimate forest age through time. Common forest disturbance detection algorithms include disturbance and trend detection (Kennedy et al., 2010), vegetation change tracker (VCT) (Huang et al., 2010), continuous change detection and classification (CCDC) (Zhu and Woodcock, 2014), and breaks for additive season and trend (Verbesselt et al., 2012; DeVries et al., 2015). Chen et al. (2016) developed the stand-replacing

fire mapping method using Landsat images from 2001 to 2012 to infer the forest age of Siberian larch. Kauffman and Prisley (2016) used the VCT algorithm to detect disturbance events based on Landsat time-series images. Diao et al. (2020) used the VCT algorithm, spatial analysis and random forest regression to map the ages of three typical plantations in southern China (1987–2017). Methods based on forest disturbance monitoring have shown strong potential for forest age estimation, but as yet there exist only a few related

studies involving large-scale mapping.

This research uses the Google Earth Engine (GEE) cloud platform with 30 m Landsat images and the CCDC algorithm to estimate forest age across the whole of China in 2020. The CCDC algorithm was selected because it can exploit the full temporal profile of long Landsat time-series data, and judge accurately the disturbance time point (Zhu and Woodcock, 2014), thereby, achieving reliable forest age mapping (Shen et al., 2018; DeVries et

al., 2015). At present, there exist very few studies mapping forest age at a fine spatial resolution and across large areas. Therefore, this study fills such a research gap by mapping forest age at 30 m spatial resolution across the whole of China. In general, the main contributions of this paper are as follows: 1) a large-scale forest age mapping method is proposed based on the CCDC algorithm, which shows potential for mapping global forest ages at the fine spatial resolution of 30 m; and 2) a 30 m spatial resolution forest age map across China in 2020,

as a preliminary result of annual forest age mapping, is produced. The dataset is available at https://doi.org/10.6084/m9.figshare.21627023.v6 for public use (Xiao, 2022).



## 2 Data

### 2.1 Landsat images

Landsat Collection 1 (C1) Tier 1 Surface Reflectance (SR) images were selected, including all available Landsat 4-8 images from 1985 to 2020. These images were obtained directly from the GEE platform (https://developers.google.com/earth-engine/datasets/catalog/landsat), with a total of 436,967 Landsat tiles across China. Furthermore, these data were atmospherically corrected using the LaSRC algorithm (Vermote et al.,

2018). We pre-processed the image within China according to the Landsat SR Quality Assessment (QA) band, including removing shadows, clouds, cloud shadows and snow-covered areas. In addition, it was necessary to remove outliers in the image; thus, pixels with reflectance less than zero in each spectral band and pixels with significantly high reflectance were removed. It should be noted that the earliest available images for each region are not the same. For example, the earliest available images in western China were significantly later than those

in the eastern coastal regions. Figure 1(a) and Figure 1(b) show the year of the earliest available Landsat 4-8 images covering China before and after masking out non-forest land, respectively. The masks used were based on the 2020 ESA land cover product (WorldCover2020). It can be seen from Figure 1(b) that the available Landsat 4-8 data after 1990 cover most of the forest land in China.

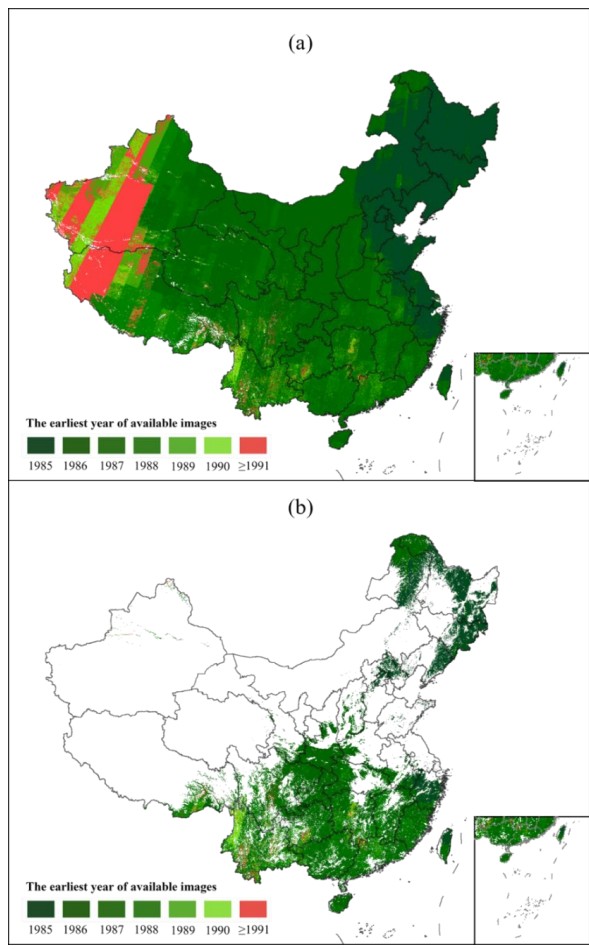

Figure 1. The earliest available year of the Landsat images used in this study. (a) Years for the whole of China. (b) Years for the non-forest areas masked out.



## 2.2 Auxiliary data

This research used several land cover products to produce reference data to calculate the stand age mapping accuracy, including the Global Forest/Non-Forest Map (FNF), Global Forest Change (GFC), Global Forest Cover Change Dataset (GFCC), Annual Global Land Cover between 2000 and 2015 (AGLC_2000_2015), Global Land Use/Land Cover Dataset (ESRIGlobal-LULC_10m) and WorldCover2020. A detailed description of these products is presented in Table 1.

Table 1 Auxiliary data used for accuracy evaluation.

| ID | LULC products | Data sources | Resolution | Selected Years | References |
|----|---------------|--------------|------------|----------------|------------|
| 1 | FNF | PALSAR-2/PALSAR | 25 m | 2010, 2015 | Shimada et al. (2014) |
| 2 | GFC | Landsat | 30 m | 2000–2020 | Hansen et al. (2013) |
| 3 | GFCC | Landsat, MODIS Vegetation Continuous Field (VCF) tree cover data | 30 m | 2000, 2005, 2010, 2015 | Sexton et al. (2013) |
| 4 | AGLC_2000_2015 | Multiple sets of global land cover products, Landsat | 30 m | 2000, 2005, 2010, 2015 | Xu et al. (2021) |
| 5 | ESRI_Global_LULC_10m | Sentinel-2 | 10 m | 2020 | Karra et al. (2021) |
| 6 | WorldCover2020 | Sentinel-1, Sentinel-2 | 10 m | 2020 | Zanaga et al. (2021) |

## 3 Methodology

### 3.1 CCDC algorithm

The CCDC algorithm is usually used to monitor land cover changes (Zhu and Woodcock, 2014; Li et al., 2021). It fits a model to spectral observations of Landsat pixels or vegetation indices (such as the normalized difference vegetation index (NDVI)), and can reflect three types of pixel changes: (1) seasonal changes (such as phenology), (2) slow changes (such as vegetation growth or degradation) and (3) rapid changes (such as deforestation, insect disasters, storms and fires) (Zhu and Woodcock, 2014). CCDC uses robust iteratively reweighted least squares (RIRLS) (Dumouchel and O'brien, 1992) to fit to the observed values, which can reflect the phenological characteristics and changing trends of ground features. The mathematical expression of the fitted line is as follows:

$$\hat{\rho}(i,x)_{RIRLS} = a_{0,i} + a_{1,i}\cos\left(\frac{2\pi}{T}x\right) + b_{1,i}\sin\left(\frac{2\pi}{T}x\right) + a_{2,i}\cos\left(\frac{2\pi}{NT}x\right) + b_{2,i}\sin\left(\frac{2\pi}{NT}x\right) \tag{1}$$

where $x$ represents Julian day, $i$ represents the $i^{th}$ band of the image, $T$ represents the number of days each year and $N$ represents the number of years of Landsat data. The coefficient $a_{0,i}$ represents the overall values of the





$i^{th}$ band, $a_{1,i}$ and $b_{1,i}$ represent the intra-annual change of the $i^{th}$ band, and $a_{2,i}$ and $b_{2,i}$ represent the inter-annual change of the $i^{th}$ band. Finally, $\hat{\rho}(i,x)_{RIRLS}$ represents the predicted value for the $i^{th}$ band corresponding to the $x^{th}$ Julian day based on RIRLS fitting.

### 3.2 Forest age mapping based on CCDC

In this research, the CCDC-based method is proposed for large-scale forest age mapping (using Landsat images from the GEE cloud platform and the CCDC algorithm). Arévalo et al. (2020) provided the CCDC application programming interface on the GEE platform so that the algorithm could be employed conveniently.

### 3.2.1 Dividing the country into small grid cells

The CCDC algorithm performs time-series analysis per-pixel, and the large-scale calculations require significant computing power. Although GEE has powerful computing capacity, it is still difficult to analyze the time-series at a national scale. For this reason, the country was divided into 62 grid cells of 5°×5° (Figure 2), as this scale not only requires less GEE computing power, but also avoids increasing data management costs.

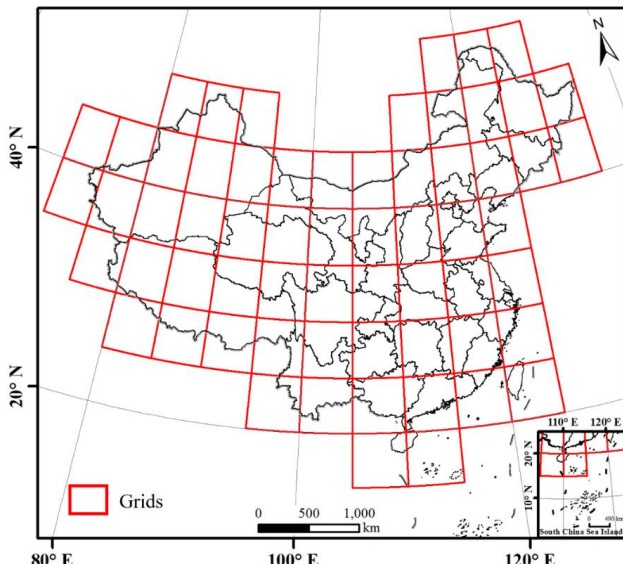

Figure 2. The divided grids (62 5°×5° grids) for the national land area.

### 3.2.2 Determining the 2020 forest distribution mask

This research utilizes existing 2020 LULC classification products to map forest distribution. Given these data, it was necessary only to identify the woodland area in the year of mapping (i.e., 2020 in this paper) and the time of the last land-replacing change in that area to estimate forest age. For example, if a pixel in the image is forestland in 2020 and the last time the area changed to forestland was in 2015, then the forestland is five years old.

Since 2020 is the target year of forest age mapping, we extracted the forest area from WorldCover2020 to generate the forest mask in 2020 (referred to as 'Forest mask 2020'). The accuracy of the WorldCover2020 forest classification is sufficient for large-scale forest age mapping (producers' accuracy and users' accuracy are 89.9% and 80.8%, respectively). In addition, the spatial resolution of WorldCover2020 is 10 m, which makes it straightforward to match with the 30 m resolution of Landsat data (i.e., 10 m WorldCover2020 data can be degraded to 30 m conveniently).

### 3.2.3 Determining the breakpoints of the model

CCDC performs time-series analyses for each pixel. The model contains two key parameters, Chi-square probability (*chiSquareProbability*) and the minimum number of consecutive observations (*minObservations*) that trigger breakpoint conditions. It should be noted that the *chiSquareProbability* value ranged from 0 to 1. The larger the parameter value, the fewer the breakpoints detected by the model. The value of *minObservations* is a positive integer, which affects the sensitivity of the algorithm to breakpoint detection. For example, if the

sensitivity is too large, then slow forest degradation (owing to insect pests and selective logging, etc.) will also be detected as breakpoints. Because there is no land cover *type* change in this process, a large sensitivity will lead to an underestimation of forest age, and *vice versa*. Therefore, finding the most suitable parameter threshold is the key to reliable forest age mapping.

### 3.2.4 Calculating the stand age

First, we determined the endpoint of the final fitted curve corresponding to each forest pixel (extracted using the Forest mask 2020). It should be noted that the Forest mask 2020 can represent only the forest extent at a certain time in 2020 and thus, this paper assumes that the Forest mask 2020 represents the forest extent on September 1, 2020 (i.e., at the end of summer characterized by green vegetation). Figure 3 shows a schematic diagram illustrating forest age determination based on time-series analysis and the Forest mask 2020. The solid

line represents the time-series fitting curve of the surface reflectance of a certain pixel, the red dotted line is the time point on September 1, 2020, and the purple curve intersecting the red dotted line indicates that the forest did not change during this period. Breakpoint $B_1$ indicates that a severe disturbance has occurred at the corresponding time. Point $C_1$ indicates that the location began to gradually recover to forest (afforestation or natural restoration) after a drastic change. Therefore, the forest age at target time point D can be estimated

from the distance of CD.

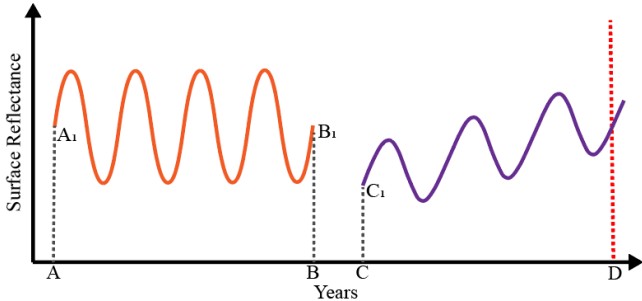

**Figure 3. Schematic diagram of forest age estimation. A1 and C1 represent the starts of the first fitting curve and second fitting curve, respectively, and B1 represents breakpoint of first disturbance. A, B, and C represent the time points of A1, B1, and C1, respectively. D indicates September 1, 2020.**

Figure 4 shows the time-series curve of a pixel analyzed by CCDC. The first row of images is the true color Landsat image at each time point centered at the pixel (red dot), and the second row is the corresponding fitting curve. The CCDC model detected two breakpoints in this pixel from 2004 to 2021. Specifically, the forest degraded slowly since 2004, and the image shows that it still belongs to woodland on June 22, 2016. After that, the model detected a breakpoint, indicating that the woodland was disturbed rapidly and the land cover type

changed. The image on August 9, 2016 shows that the location was covered by bare land at this time, and after a period of restoration, vegetation began to regrow. The images on August 7 and August 28, 2018 show that it was fully restored to woodland finally.



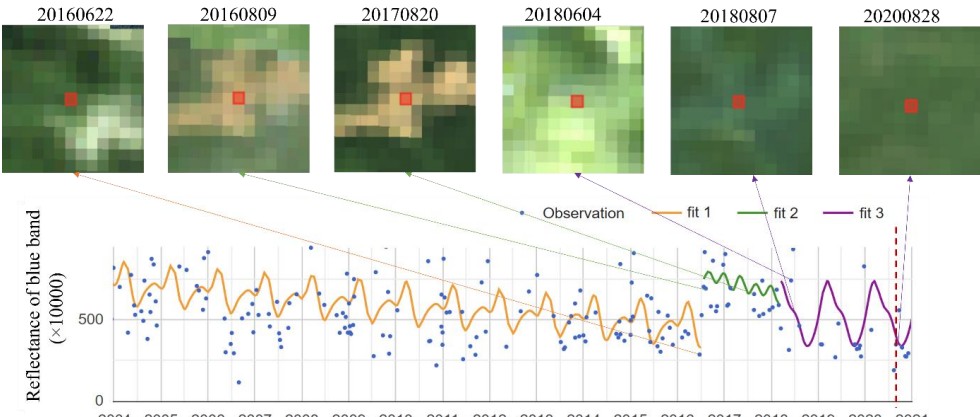

Figure 4. Time-series analysis of a pixel by CCDC (only shows blue band of Landsat images).

### 3.3 Generation of validation samples using LULC products

Accuracy assessment for large areas generally requires a large number of validation samples. The existing large-scale validation sample sets often contain data in a certain year only, but accuracy evaluation of forest age requires multi-period sample sets. Currently, many LULC products are available, and researchers have invested considerable work in ensuring product accuracy. Therefore, this research used comprehensively these LULC products to generate validation samples. To ensure the reliability of the samples, only land cover products after 2000 were used because there were few existing land cover products before 2000.

The 1–20 year stand age was grouped into four stand age classes: 16–20 years, 11–15 years, 6–10 years, and 1–5 years. These were then converted into binary classification maps with two classes: regrowth and non-regrowth. For the reference data, we used the LULC products to generate regrowth and non-regrowth samples every five years after 2000. Regrowth samples from 2000–2005, 2006–2010, 2011–2015, and 2016–2020 were used to create four stand age classes; 16–20 years, 11–15 years, 6–10 years, and 1–5 years, respectively. If they are unified, then the predicted age of the pixel is considered correct; otherwise, it is considered as misclassified. Figure 5 is a flowchart showing how the LULC products were used to generate the validation samples. The following section introduces explicitly the accuracy evaluation process.

(1) *Extracting forest areas of selected years from LULC products*. Because the available years for each product are not uniform, several years were selected from the available years, with multiple products at the same time in these years normally available. We identified five years: 2000, 2005, 2010, 2015, and 2020. The forest mask (FM) for these five years was first extracted from the LULC products. To ensure the reliability of the sample, the intersection of the FM of different LULC products each year (areas that were classified as forest by all LULC products) was determined, and the intersection area was considered as the consensus forest (CF), while areas that were classified as forest by only one product were designated as undefined forest (UF).

(2) *Differencing*. Differencing of the FMs of the years before and after each period was performed to assess the consensus regrowth (CR) in the four periods, that is, 2000–2005, 2005–2010, 2010–2015 and 2015–2020. Since UF cannot determine whether it is forest, the UF of the years before and after each period does not participate in the differencing process. The union of these two areas was defined as undefined regrowth (UR). The area remaining in the image after removing the CR and UR was defined as consensus non-regrowth (CN). Specifically, UR, CR and CN are expressed as follows:

$$\begin{cases} UR = UF_{t_1} \bigcup UF_{t_2} \\ CR = CF_{t_2} - CF_{t_1} - UR \\ CN = I - UR - CR \end{cases} \quad (2)$$

where *CR*, *UR*, *CF*, *UF* and *CN* represent the spatial sets of CR, UR, CF, UF and CN, respectively, *I* represents the





265   spatial set of the entire image area, $t_1$ and $t_2$ represent the two years before and after each period, respectively, and $\cup$ represents the union of the sets.

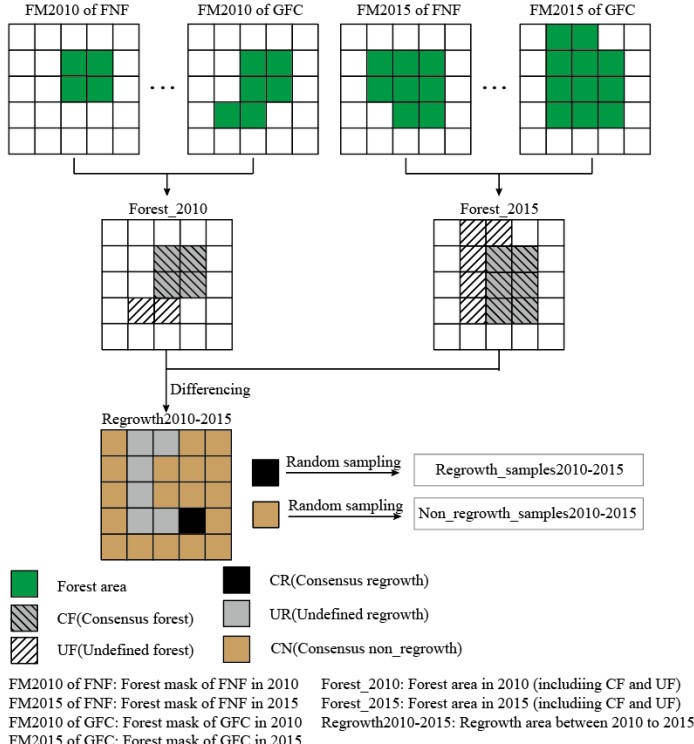

Figure 5. Validation samples generated using LULC products.

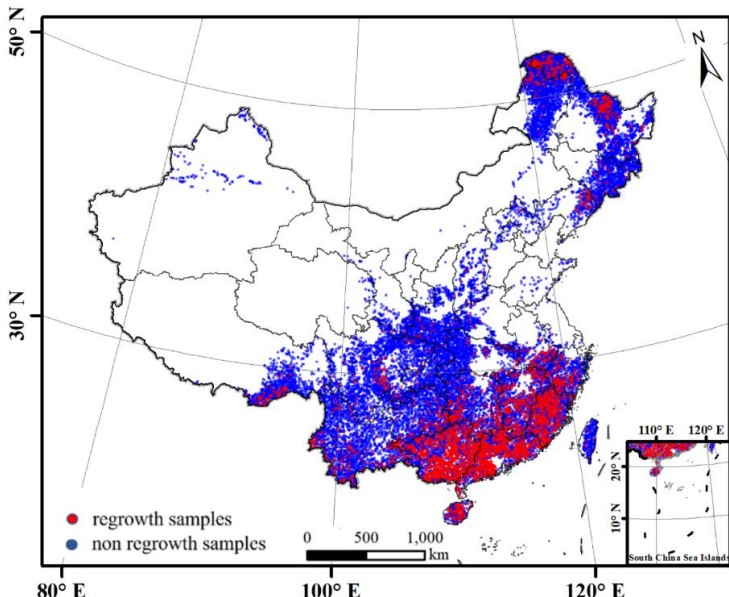

270   Figure 6. Distribution map of validation samples.

(3) *Random sampling and confusion matrix calculation.* Stratified random sampling was used to generate



validation sample sets containing 1,000 regrowth and 5,000 non-regrowth samples for each period of CR and UR. Considering the possibility of regrowth events occurring in each period within the same pixel, only the regrowth samples in the most recent period were retained for the regrowth samples in the four periods. As a result, 2,618 regrowth samples (red dots in Figure 6) and 21,007 non-new forestland samples (blue dots in Figure 6) were obtained.

## 4 Results

### 4.1 Validation of the produced forest age map

#### 4.1.1 National- and provincial-Level Performance

The validation samples (reference data) in each period and the forest age predicted by the model (predicted data) were compared to form a confusion matrix. The overall accuracy (OA) of the national young forest age mapping was found to be 90.28% (Table 2). In addition, this research considered the cartographic performance of the proposed method in various provinces in China (Figure 7). To ensure consistency in the number of samples used, the number of regrowth and non-regrowth samples for each province was controlled at around 400. In general, the OA of young forest age mapping in all provinces in China was larger than 54%, and the OAs of Ningxia, Macau, Tianjin, Fujian, Zhejiang, Anhui, and Guangdong were all larger than 80%. Except for Ningxia, the other six provinces (cities and autonomous regions) are located in eastern and southern China. The provinces with relatively weak classification performance were Gansu, Jiangxi, Shaanxi and Beijing (in order), and the OAs of these four provinces were smaller than 60%. Except for the above provinces, the OAs of the remaining provinces were between 60% and 80%. In general, the classification performance of the southern provinces was more accurate than that of the northern provinces.

Table 2. Confusion matrix of regrowth and non-regrowth.

|  |  | Predicted data | | | |
|---|---|---|---|---|---|
|  |  | Non-regrowth | Regrowth | Total | Producers' Accuracy (%) |
|  | Non-regrowth | 19,299 | 589 | 19,888 | 97.04 |
|  | Regrowth | 1,708 | 2,029 | 3,737 | •54.29 |
| Reference data | Total | 21,007 | 2,618 | 23,625 |  |
|  | Users' Accuracy (%) | 91.87 | 77.50 |  |  |
|  |  |  |  |  | Overall Accuracy: 90.28 % |

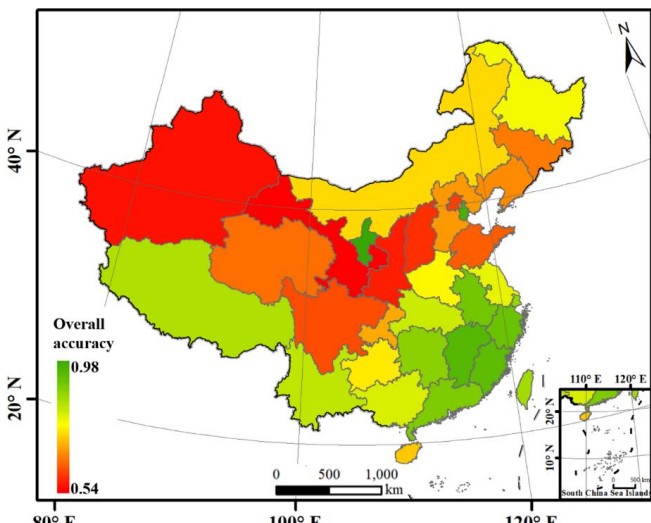

Figure 7. Overall accuracy of young forest age mapping across different provinces in China.

## 4.1.2 Comparison with existing products

We compared visually the forest age map produced by proposed method with the MPI-BGC forest age dataset (at 1 km spatial resolution) (Besnard et al., 2021). Figure 8 shows three cases for comparison. In case 1, MPI-BGC presents much less information on the forest age compared to proposed method. The reason may be that
MPI-BGC is produced based on the relationship between forest age and forest biomass, which is influenced greatly by different forest types. However, this research estimates forest age based on the history of forest disturbance and, thus, is not affected by the forest type. Moreover, there are more age classes mixed within the area of each 1 km pixel, and the MPI-BGC forest age dataset cannot present the information explicitly. In case 2, MPI-BGC depicted only the forest age in the north part of the region. It is difficult for MPI-BGC to map the age
of small-scale forests in the south part because of the coarse spatial resolution (i.e., the small-scale forests were incorrectly identified as non-forests in the 1 km data). In case 3, we selected an area dominated by small-scale forests. It is seen that MPI-BGC cannot depict the age of these forests. On the contrary, the forest age map produced by this research presents clear information at the 30 m spatial resolution, which is likely to be more helpful for monitoring small-scale deforestation activities and estimating land-atmosphere carbon fluxes than
products with a coarse spatial resolution (e.g., 1 km in the MPI-BGC forest age dataset here).
To further examine the reliability of the forest age map produced by this research, Pearson's product-moment correlation coefficient was calculated between the predicted years of regrowth and years of forest loss extracted from Hansen's product (FLH). FLH was chosen to compare with the forest age map produced by this research, as forest age products with the same time range and spatial resolution are not available. However, the FLH depicts
the distribution of annual forest loss at the global scale with a spatial resolution of 30 m from 2000 to 2020. Generally, forest regrowth occurs during the recovery phase after forest loss. Therefore, the soundness of the proposed method can be reflected to some extent by this Pearson's product-moment correlation analysis. Specifically, after 2000, 10,000 samples were selected randomly from the regrowth areas in the country. The results showed that there was a large correlation between the years of forest regrowth predicted by this
research and the years of FLH, with a Pearson's correlation coefficient of 0.62. As shown in Figure 9, a large number of sample points were distributed on the diagonal line ($y = x$) or near the right side because the forest at these observation points could be quickly restored to forest after being disturbed. At the same time, the point density in the lower-right part of the diagonal is significantly larger than that in the upper-left part, indicating that the forest age estimation for most of the sample points is reasonable. Observations in the upper left part of





the diagonal line represent areas where forest age may be underestimated or misclassified as forest loss from the FLH.

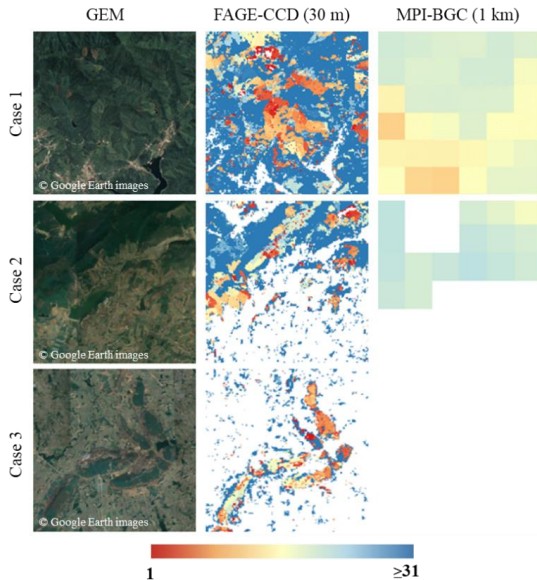

**Figure 8. Three scenarios for comparison between the 30 m spatial resolution product (based on CCDC) and the 1 km spatial resolution product (based on MPI-BGC). White pixels of the forest age maps in the second and third columns**
**indicate non-forest or no data.**

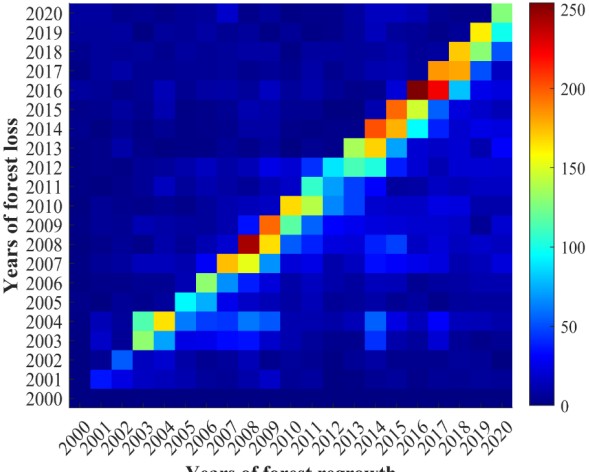

**Figure 9. Years of predicted regrowth versus years of FLH after 2000. The value of the color bar represents the number of samples that fall within each pixel.**

### 4.1.3 Evaluation based on field measurements

We searched 150 relevant papers published after 2020 from China National Knowledge Infrastructure (CNKI) using the following keywords: China and forest age. Finally, 23 field measurements with accurate geographic and forest age were generated after filtering out sites that do not overlap with the 2020 forest mask and sites that are older than 31 years, see Table 3. Figure 10 shows the scatter-plot between the field measurements and



predicted forest age. Referring to the field measurements, the predicted forest age has a correlation coefficient
of 0.81 and root mean square error (RMSE) of 5.58, suggesting an acceptable correlation with the field
measurements.

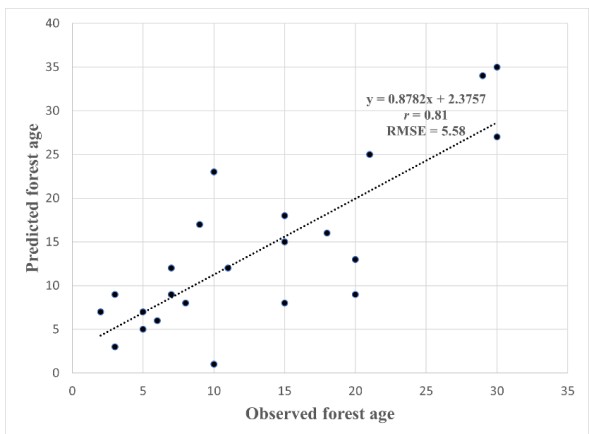

Figure 10. Comparison between the forest age derived from field measurements (observed forest age) and predicted forest
age.

**Table 3. Information on the 23 field measurements.**

| ID | Longitude | Latitude | Observed forest age | Source |
|---|---|---|---|---|
| 1 | 109.328858 | 23.050233 | 3 | Li et al. (2021) |
| 2 | 109.332939 | 23.053525 | 8 | Li et al. (2021) |
| 3 | 109.242036 | 23.111756 | 18 | Li et al. (2021) |
| 4 | 109.160242 | 23.053275 | 21 | Li et al. (2021) |
| 5 | 109.159194 | 23.040914 | 29 | Li et al. (2021) |
| 6 | 122.491287 | 42.717326 | 20 | Han et al. (2022) |
| 7 | 122.571380 | 42.684847 | 30 | Han et al. (2022) |
| 8 | 113.421000 | 23.245000 | 6 | Chen et al. (2022) |
| 9 | 113.393000 | 23.226000 | 10 | Chen et al. (2022) |
| 10 | 113.419000 | 23.256000 | 15 | Chen et al. (2022) |
| 11 | 113.394000 | 23.212000 | 20 | Chen et al. (2022) |
| 12 | 113.381000 | 23.255000 | 30 | Chen et al. (2022) |
| 13 | 106.740000 | 26.520000 | 11 | Yin et al. (2021) |
| 14 | 110.465833 | 22.048333 | 5 | Song et al. (2021) |
| 15 | 110.500833 | 21.919167 | 15 | Song et al. (2021) |
| 16 | 110.500278 | 22.022222 | 5 | Song et al. (2021) |
| 17 | 110.517500 | 21.908056 | 15 | Song et al. (2021) |
| 18 | 110.516111 | 21.908056 | 10 | Song et al. (2021) |
| 19 | 117.935278 | 26.881389 | 7 | Feng et al. (2021) |
| 20 | 118.451667 | 26.243333 | 2 | Hong et al. (2021) |
| 21 | 116.650833 | 25.172778 | 3 | Hong et al. (2021) |
| 22 | 118.351389 | 27.317500 | 7 | Hong et al. (2021) |
| 23 | 117.802222 | 27.275556 | 9 | Hong et al. (2021) |



### 4.2 Analysis of key parameters in CCDC

The sensitivity of the model to breakpoint detection affects directly the accuracy of stand age mapping, and the two parameters *chiSquareProbability* and *minObservations* play important roles in the model. To determine the optimal parameters, we selected five regions in China (Figure 11) for testing. These five regions are all sized
0.5°×0.5° and distributed in the northeast, southwest, central and eastern regions of China. In this research, the value of the *chiSquareProbability* parameter was increased from 0.50 to 0.99, while *minObservations* was increased from 2 to 20.

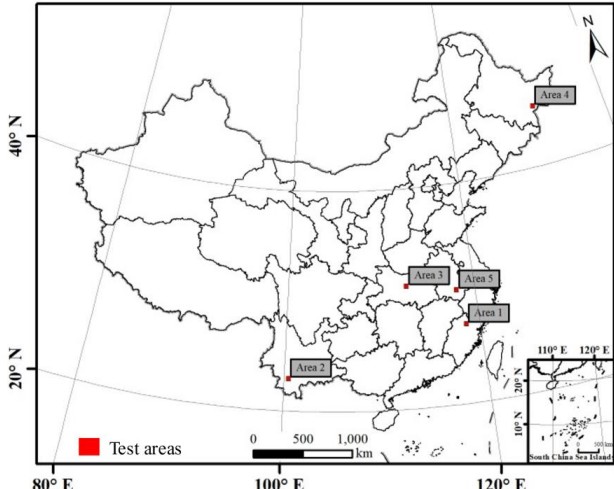

Figure 11. Spatial distribution of the five test areas for analyzing the influence of key parameters.

### 4.2.1 Analysis of *chiSquareProbability*

Figure 12(a) shows that the OA of stand age mapping in the five areas varies with the choice of different *chiSquareProbability* values. The largest OAs of the other four areas except Area 3 occur when the *chiSquareProbability* value is around 0.98, whereas the largest OA of Area 3 occurs when the *chiSquareProbability* value is 0.82. The OA of Area 3 reaches the largest value earlier, as the forest land in this
area is disturbed more frequently. In this case, the CCDC model requires a smaller *chiSquareProbability* value to detect more breakpoints. In addition, Figure 12(a) shows that the OA increase in Area 2 is the fastest, with the smallest OA (70.16%) observed when the *chiSquareProbability* value is 0.50 and the largest (90.35%) observed when the *chiSquareProbability* value is 0.99. The largest and smallest OA presented a difference of 20.19%. The reason for this phenomenon may be that the disturbance year of the forest in Area 2 was relatively late and the
forest experienced less disturbance. When the *chiSquareProbability* value is too small, more breakpoints will be detected incorrectly, which affects the OA of the forest age mapping.





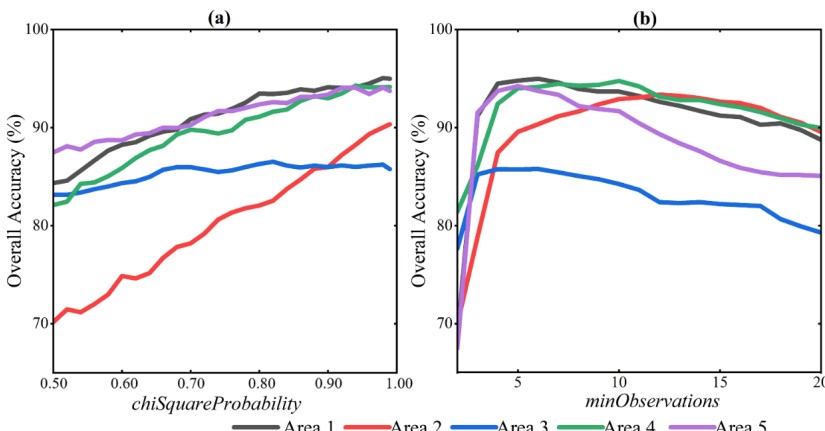

**Figure 12. OA of forest age under different values of (a) chiSquareProbability and (b) minObservations in five regions.**

Figure 13 shows the model performance when different *chiSquareProbability* values were used. Specifically,
columns 1, 2, and 3 show the stand age maps of the five regions when the parameter *chiSquareProbability* values are 0.50, 0.74 and 0.99, respectively. As the value of the parameter *chiSquareProbability* increases, the area of regrowth detected by the CCDC algorithm decreases. When the value was 0.50, the stand age map for each region contains a large number of misclassified regrowth areas. These misclassified regrowth areas are due mainly to the small values of *chiSquareProbability*, which make the model extremely sensitive to breakpoint
detection.

Generally, there is a close relationship between forest restoration and forest loss. For this reason, FLH was added to the fourth column for convenient visual comparison. The color of the FLH indicates the year of forest loss. As the earliest available year for FLH is 2000, the fourth column of Figure 13 shows only the years of forest loss after 2000. The fifth column of Figure 13 shows the corresponding fine spatial resolution Google Earth maps (GEMs).
Clear traces of forest disturbance can be observed in the five regions from the GEMs. These areas are more consistent with the dark red areas in the third column of the stand age maps.

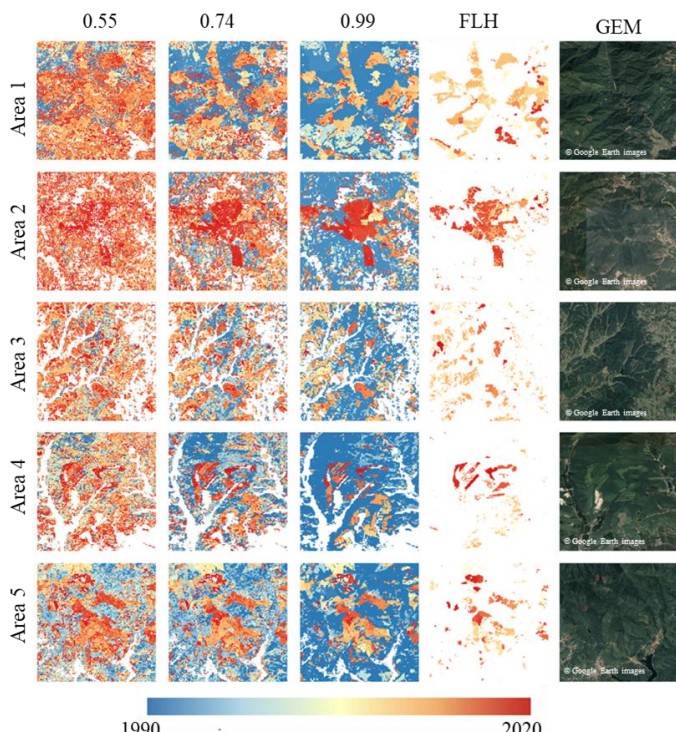

**Figure 13. Stand age maps of the five regions (marked in Figure 11) under different values of chiSquareProbability (0.55, 0.74, and 0.99).**

### 4.2.2 Analysis of *minObservations*

Figure 12(b) shows that the OA of stand age in the five regions varied with *minObservations*. The OAs of stand age in the five areas show a trend of initially increasing and then decreasing. This means that when the *minObservations* value is smaller, the CCDC model can detect more breakpoints while producing more misclassified regrowth values. When the *minObservations* value exceeds the optimal threshold, the model presents incorrect detection results. When the parameter is less than six, the OAs of the five regions increase rapidly. When the parameter is greater than 12, the OAs of each region enter a stage of rapid decay. The largest OAs for both Area 1 (94.98%) and Area 3 (85.78%) occur when the values of *minObservations* are equal to six. The OA of Area 5 reaches the maximum value (94.23%) when *minObservations* is five, while Area 4 and Area 2 reach the maximum OA (94.75% and 93.37%, respectively) when the values of *minObservations* are 10 and 12, respectively.

### 4.3 Spatial distribution of young forests in China

This research produced a young forest stand age map in China in 2020, with a spatial resolution of 30 m (Figure 14(a)). To show the spatial distribution of young forest age more clearly, we divided the forest into four stand age classes, namely stand age class I (1–10 years), II (11–20 years), III (21–31 years) and IV (> 31 years). The stand age class IV accounted for 81% of the total forest area in 2020, while the other three stand age classes accounted for 19%. In the young forests, stand age class III accounted for the largest proportion (39.32%), followed by stand age class II (38.34%). Stand age class I (22.34%) accounted for the smallest proportion. This means that the country experienced many afforestation and/or reforestation events from 1990 to 2000, and the

speed of this process slowed after 2000. The main reason may be that the country's early policies (specifically, the Returning Farmland to Forest Program and the Afforestation Program) were implemented effectively, and by 2000 many areas suitable for afforestation had been occupied.

Young forestland in China is distributed mainly in the southern provinces of China, such as Yunnan, Guangxi, Guangdong and Fujian. As these provinces are located in a subtropical climate zone, abundant rainfall and suitable climatic conditions make them suitable for tree growth. In addition, Figure 14(c) shows that there is more young regrowth in the Daxing'anling region of northeastern Inner Mongolia, partly because of the large possibility of forest fires in the virgin forests in this area, and large areas of forest have recovered to young regrowth after fire disturbance (Zhang et al., 2017). In addition, we found that this area is characterized by long snow accumulation periods and large mountain slopes; therefore, many pixels in this area were misclassified as young regrowth. In general, the growth rate of young regrowth in China showed a decreasing trend during the study period (1990–2020), indicating a decrease in the area available for afforestation.

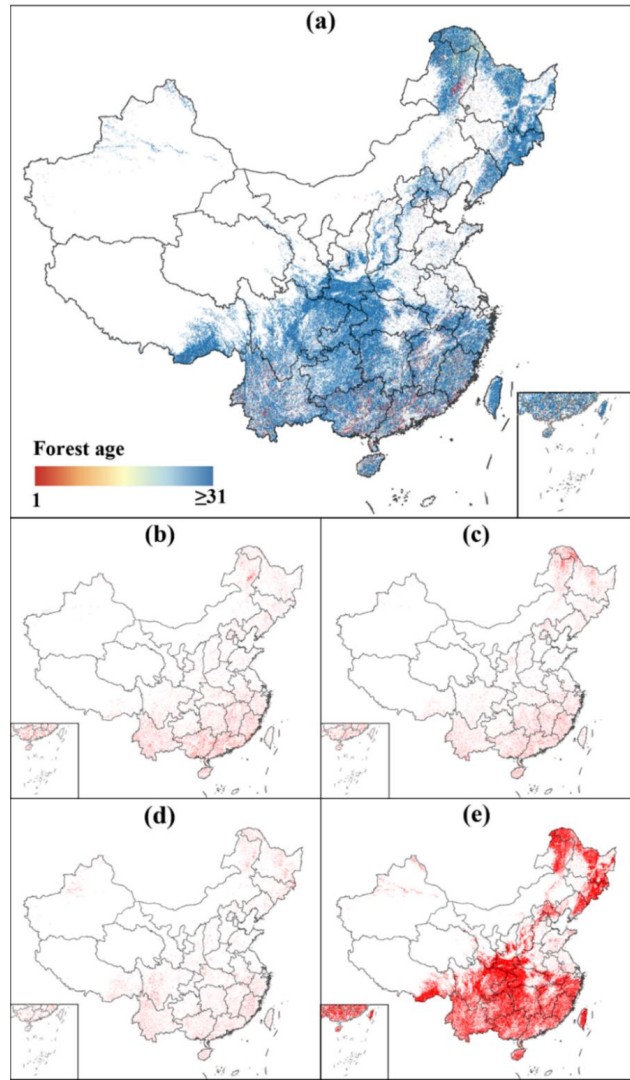

Figure 14. Stand age map in China at 30 m spatial resolution. (a) Chinese stand ages and stand age classes, (b) 1–10 years, (c) 11–20 years, (d) 21–31 years, and (e) >31years.



### 4.4 Average age of young forests in different provinces

Figure 15 shows the average age distribution of young forests across the provinces of China. Interestingly, the age is larger in the north than the south, and larger in the west than the east. This phenomenon is driven mainly by natural and anthropogenic factors. Generally, tree growth in western China is restricted by the natural environment. The fragile ecological environment forces people to protect forests in this area, and the proportion of economically productive forests is small. Moreover, the Three-North Shelter Forest Program, which began in

1978, has enabled the effective protection of forestland in the northern region (Wang et al., 2007; Qiu et al., 2017). Therefore, the average forest age in the west is relatively large. On the other hand, a large number of economically productive forests were used to provide timber in southern China, leading to young forest regrowth in the south. Therefore, the average forest age is smaller. In addition, forests have experienced more disturbance due to rapid urban expansion in eastern and southern China (Meng et al., 2020).

The average age of young forests in each province was ranked in ascending order, with Tianjin, Guangxi, Shandong, and Guangdong ranking first (11.3 years), second (11.7 years), third (11.9 years) and fourth (12.2 years), respectively. These provinces are located in southern and eastern China. Furthermore, the average age of young forests in the Ningxia Hui Autonomous Region is relatively young and ranks fifth (12.6 years) as the forest resources of the Ningxia Hui Autonomous Region have further increased in the past 30 years based on the

Returning Farmland to Forest Program, the Afforestation Program and the Three-North Shelter Forest Program (Wang et al., 2007; Qiu et al., 2017)(Wang et al., 2007; Qiu et al., 2017). When the average age of young forests in each province was ranged in descending order, the top five provinces (cities and autonomous regions) are Xinjiang (25.7 years), Hong Kong (20.3 years), Tibet (19.5 years), Qinghai (18.9 years), Sichuan (18.6 years) and Shaanxi (18.3 years). Except for the Hong Kong Special Administrative Region, the other four provinces are all in

the western region because the special natural conditions in western China make afforestation or natural restoration of forests difficult. The average age of young forests in Hong Kong is relatively large because of the limited afforestation in the area. Therefore, to further strengthen the role of China's young forest lands in the "carbon neutrality" initiative, it is particularly important to carry out afforestation suitability assessments in China (especially in the western and northwestern regions) (Zhang et al., 2022).


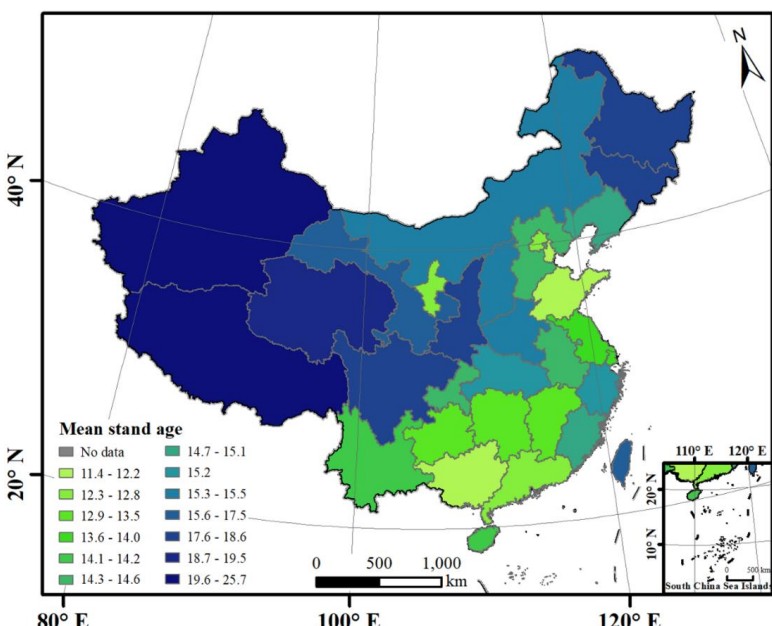

**Figure 15. The average age of young forests in various provinces in China.**



## 5 Discussion

### 5.1 Effect of input features on the model

Several studies have used the normalized degradation fraction index (NDFI) to increase the accuracy of forest disturbance detection (Souza et al., 2005; Bullock et al., 2020; Chen et al., 2021). The NDFI is calculated from the abundance of several endmembers, including soil, shadow, green vegetation (GV) and non-photosynthetic vegetation (NPV), through spectral unmixing. To explore the influence of different features on forest age mapping, this research first set the two parameters of *chiSquareProbability* and *minObservations* to 0.99 and 6,

respectively, and then input the following different features to the CCDC model: spectral bands of Landsat images (spectral), abundance of four endmembers (GV, Shade, NPV and Soil) and index features (NDFI, NDVI, normalized burning index (NBR), normalized difference moisture index (NDMI) and enhanced vegetation index (EVI)). The steps of spectral unmixing were described by Chen et al. (2021).

Figure 16 shows the OAs of the five regions with the input of different features. Using the original Landsat bands

as the input to the model can achieve the greatest mapping accuracy. Except for the spectral feature, whose performance is relatively stable in the five regions, the performance of the other features in the five regions is quite different. For example, in Area 1, the mapping performance of the NDFI-based feature is the most satisfactory (the OA is 90.29%), and the performance of the GV-based feature is the weakest (76.00%); in Area 2, the performance of the GV-based feature is the most satisfactory (the OA = 82.28 %), and the performance of

the soil-based feature is the weakest (the OA is 71.85%). Generally, EVI (71.83%), EVI/NDVI (82.43%) and EVI (60.07%) were the least predictive features in these three regions.

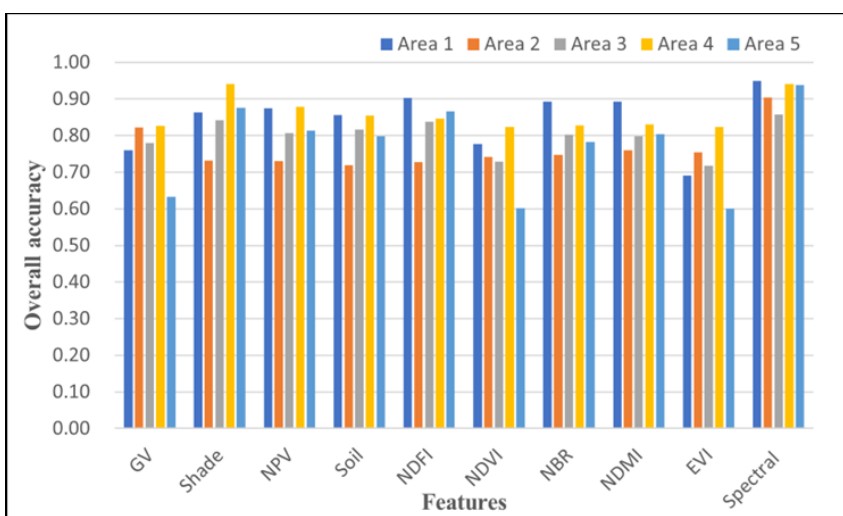

**Figure 16. OA of the CCDC-based method with different input features in five regions.**

### 5.2 Whether to choose vegetation growing season images

To eliminate the influence of winter ice and snow and improve model fitting, images of the peak vegetation growth season in a year are often selected as observation data, such as Landsat images from the 150[th] to 300[th] day of each year (Chen et al. 2021). However, this method of selecting parts of images of the year reduces the available information, especially in warmer regions (where snow and ice are short-lived or largely unaffected by snow and ice). This research compared the mapping accuracy when all the images and some images (the

images of the 150[th] to 300[th] day of each year) were selected from the annual images as the model input. The OA of young forest stand age mapping using partial data as model input was 88.53%. When using partial images,



the OA of the national young forest age mapping was 1.75% smaller than that when using all the images (90.28%).

To further explore the mapping differences between the two input strategies, the difference in the OA for each
province was calculated, as shown in Figure 17. Except for Tianjin, the Ningxia Hui Autonomous Region, Heilongjiang, Jilin and Qinghai, the OAs of using partial data in the other 27 provinces (cities and autonomous regions) are smaller than that of using all data. Among the 27 provinces (cities and autonomous regions), Tibet, Yunnan and Guangdong show large differences, with differences in OA ranging from 14.93% to 19.69%, followed by Guangxi, Jiangsu, Shanghai, Henan, Fujian, Anhui, Hunan and Hong Kong (OA differences between 14.93%
and 4.70%). Except for the abovementioned provinces (cities and autonomous regions), the OAs of the remaining provinces (cities and autonomous regions) are within a 4.70% difference. The above comparison shows that the use of partial image sets generally reduces the mapping accuracy in most areas.

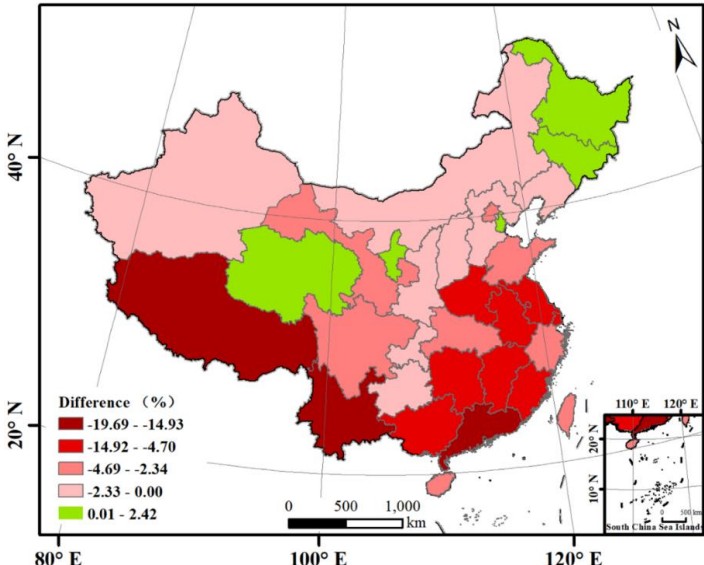

Figure 17. Difference in OA (in units of province) between the use of partial and full data.

**5.3 Application potential of the proposed method**

This research used 436,967 Landsat tiles across China to map forest age at a spatial resolution of 30 m, which validated the feasibility of the proposed method for "big" data processing. In future, the model can be used to generate a global-scale young forest age dataset. This dataset will help build a global-scale forest carbon cycle model and potentially increase the estimation accuracy of carbon sources and sinks (Wang et al., 2020; Piao et
al., 2022). In addition, studies have shown that multi-aged stands have stronger carbon sink recovery ability after disturbance than even-aged stands (Tang et al., 2017); therefore, fine spatial resolution stand age datasets can be used to study the carbon sink potential of two types of stands at the global scale after disturbance.

This research not only provides basic scientific data for researchers, but also provides important references for policymakers and forest managers. Previous studies have shown that young forests have certain advantages in
carbon sequestration, but are weak in ecosystem services (Jonsson et al., 2020). That is, old forests are still irreplaceable in terms of services such as maintaining species diversity (Betts et al., 2022). Therefore, it is also necessary to maintain ecosystem services while increasing the carbon sequestration capacity of forest ecosystems under a climate change environment. The proposed CCDC-based method can estimate young forest age in real time and, thus, has the potential to be applied for dynamic monitoring of stand age structure,
such as timely detection of forest age structure and prevention of rapid forest rejuvenation.





### 5.4 Uncertainty analysis

This research uses WorldCover2020 to determine the forest distribution; however, the classification process used for its products has certain uncertainties. Specifically, the data represent the state of forest cover in 2020 rather than the cover at a certain time of the year. Therefore, this paper assumed that WorldCover2020

represents the state of forest cover on September 1, 2020, which may lead to uncertainty, mainly for areas where forest disturbances occurred in 2020. The accuracy analysis of different provinces shows that the proposed method exhibits obvious differences in performance between different provinces. The reason may be that the forests in different regions have different climatic conditions and geographical environments (such as topography, slope, altitude, etc.). This uncertainty also exists in the process of current studies that estimate stand

age using the relationships between height-age and biomass-age (Zhang et al., 2014; Zhang et al., 2017). Different disturbance frequencies also have a certain impact on the model. For example, forest succession is faster in southern China (high disturbance frequency), but relatively slow in western and northeastern China (low disturbance frequency). Therefore, the value of *chiSquareProbability* and *minObservations* of should be controlled adaptively for different forest disturbance frequencies.

This research predicted the annual forest age across China. However, it is difficult to validate the produced forest age at the temporal resolution of one year due to the lack of reference data. In this paper, coarse forest age classes (with 5 years intervals) were created to match the validation set by integrating multiple LULC products, which brings uncertainty in assessing the accuracy of the produced maps. In general, if forest age classes with finer temporal resolution are created, the accuracy is likely to be greater. However, a sufficient

number of LULC products are needed to ensure the reliability of the reference data. Thus, it is necessary to maintain the balance between the temporal resolution of forest age classes and the number of LULC products. In future, it will be of great interest to evaluate the performance of the produced dataset using age classes with finer temporal resolution, if the appropriate validation sets become available.

### 6 Data availability

The produced 30 m map of young forest age across China in this research is openly available at https://doi.org/10.6084/m9.figshare.21627023.v6 (Xiao, 2022). The Landsat data and the auxiliary data are from public data archive and user team of GEE (https://code.earthengine.google.com/).

### 7 Conclusion

Mapping the age of young forest stands is of great significance for China's strategic target of "carbon neutrality".

Conventional stand age mapping methods rely heavily on forest inventory data, but the existing forest inventory data in China are difficult to obtain and updated slowly. Moreover, the existing stand age products in China derived from remote sensing images are of coarse spatial resolution, which cannot meet the needs of stand calculations at the regional scale. In this research, we analyzed Landsat time-series images based on the CCDC model to produce a map of young stand age across the whole of China at 30 m spatial resolution. The

advantage of the mapping method is that it does not rely on forest inventory data and enables rapid mapping of young forests on a global scale using the GEE platform. The results showed that the OA of the generated map of young stand age across China was 90.28%. This dataset is significant for studying the ecosystem services and carbon cycles of young forests in China. The proposed CCDC-based method can be extended in future to global mapping of young forests.

**Author contributions**





YX designed the research, analyzed the data, wrote the original manuscript, and produced the dataset. QW revised the whole manuscript and provided the funding to support the research. XT and PMA provided direction and comments. All authors edited and approved the final manuscript.

**Competing interests**

The authors declare that they have no conflict of interest.

**Acknowledgment**

This research was supported by the National Natural Science Foundation of China under Grants 42222108, 42171345, 41971297, and 42221002.

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
