# Peer review of "m Map of Young Forest Age in China"

_Earth System Science Data, 2022_

## Referee Comment (RC1)

This study developed a 30 m young forest age map in China using Landsat images covering the period of 1990 to 2020. I found this study quite interesting and I like this idea. The approach used was straightforward and well validated. I have two major suggestions. First, a substantial amount of the contents in sections 4.3&4.4 are discussion. Suggest to reorganize these parts. Second, it's good to see the sensitivity test in this study (the analysis of key parameters in section 4.2). This help strengthen the validity of the parameters used, but this is always tricky for spatial data. My concern is that China's forests are greatly varied and how to validate that the five areas chose are representative? Other minor suggestions:

1.  In Table1, the author listed all the gridded data used. I think the forest definitions might differ between these datasets. Have you consider the definition differences and how you deal with this issue? Does it affect the results?
2.  Line205, 'too large' -> 'too high'. Same to Line206, because sensitivity should be described by high/low.
3.  For figure5, why the second column forest grids were all classified as UF? For 2015, forest was identified in the second column of the 2nd and 3rd rows of the both datasets. Seems these two grids are also regrowth forests according to the classifier defined.
    Also, there is a typo of 'includiing', which shoud be 'including'.
4.  Line272, I am not very clear how the validation sample sets were generated. Could you provide more information here?
5.  Line289, 'smaller'->'lower'
6.  Line308-309, better rephrase this sentence: "more ... than ...". It is not appropriate to compare these two since your data only limited to young forests, while MPI-BGC covers all ages.
7.  Line318, why randomly selected samples but not all the regrowth data?
8.  Lines403-404, This may not be the case. For example, it could be the reason that the forestation areas remained the same but the forest establishment (tree survival rate) was lower in recent decade. To make this claim, you need to refer to the data of forestry yearbook.
9.  Line426-428, Yes, this is reasonable. Suggest to use eucalyptus, which has been widely planted in Guangxi and Guangdong, as an example. Eucalyptus is a fast growing species and is generally harvested in 5-10 years.

---

## Community Comment (CC2)

High-resolution forest age mapping is an important part of carbon cycle research and is one of the most significant research points. Based on the CCDC algorithm, this study maps the age of young forests in China with a resolution of 30 meters. This product are valuable for the calculation of the carbon cycle and carbon budget. As a user, I am very interested in your dataset, but I found some limitations in this dataset which may hinder its further applications.

First, it is found that there are serious spatial discontinuity in this dataset, such as the following regions: R1 (121-122°E, 50-51°N), R2 (123-125°E, 51-52°N), R3 (117-119°E, 29-30°N), R4 (119-120°E, 28-29°N).

[Figure]

R1

[Figure]

[Figure]

R2

[Figure]

R3

R4

Second, the forest age mapping was carried out based on the CCDC algorithm, but it has been demonstrated that the CCDC algorithm had several limitations. (1) It did not consider the spatial differences between pixels (Ye et. al., 2023). (2) It did not consider the varied temporal consistency of the Landsat time series (Zhu et. al., 2020). (3) Large inconsistency of disturbance maps existed between the adjacent Landsat path overlap and non-overlap regions (Qiu et. al., 2022 Characterization of land disturbances based on Landsat time series). Why not use a better version of the CCDC-family algorithms such as COLD (Zhu et. al., 2020), Bi-CCD (Zheng et. al., 2021), S-

CCD (Ye et. al., 2021), NRT-MONITOR (Shang et. al., 2022), OB-COLD (Ye et. al., 2023)?

Last, it was reported that young forests under 31 years old account for 19% of China's total forest (Lines 400-401, page 16), which is quite different from the results of the ninth forest inventory in China. These differences should be clearly explained.

---

## Author Response (AR2)

Dear Editor and Reviewers,

Thank you very much for taking time out of your busy schedule to improve our manuscript. We have carefully considered your comments and revised the paper accordingly. The revised parts are marked in yellow. The main correction in the paper and the responses to the reviewers' comments are as flowing:

Response to referee #1:
This study developed a 30 m young forest age map in China using Landsat images covering the period of 1990 to 2020. I found this study quite interesting and I like this idea. The approach used was straightforward and well validated. I have two major suggestions. First, a substantial amount of the contents in sections 4.3&4.4 are discussion. Suggest to reorganize these parts. Second, it's good to see the sensitivity test in this study (the analysis of key parameters in section 4.2). This help strengthen the validity of the parameters used, but this is always tricky for spatial data. My concern is that China's forests are greatly varied and how to validate that the five areas chose are representative?

Response:

Thank you very much for your positive comments and constructive suggestions. We have carefully considered your comments and have responded as follows.

First, Sections 4.3&4.4 in this manuscript have been moved to Sections 5.1&5.2.

*"5 Discussion*
*5.1 Spatial distribution of young forests in China*
*5.2 Average age of young forests in different provinces*
*5.3 ..."*

Second, in the original version, we selected five areas to confirm the threshold of key parameters for estimating forest age using the CCDC algorithm. These five regions come from the east (Area 1 and Area 5), southwest (Area 2), central (Area 3), and northeast (Area 4) of China, and the forests in these areas account for the main part of forests in China. As you said, however, it is possible that these five areas do not completely represent the forest characteristics of the entire China. To this end, **we added other three test areas in the new version**, they are distributed in the northwest (Area 6), north (Area 7), and south (Area 8) of China. These eight test areas come from seven major geographical regions of China, which have different geographical and vegetation growth conditions that cannot be ignored in forest age mapping. In addition, these eight areas also represent different forest protection policies and forestry uses. For example, the forests in the Three-North Shelter Forest Region are mainly protected, while there are a large number of plantation forests for timber production in southern China. The corresponding figures and texts have been

modified as follows.

*"4.2 Analysis of key parameters in CCDC*

[revised manuscript text omitted]

Other minor suggestions:
1. In Table1, the author listed all the gridded data used. I think the forest definitions might differ between these datasets. Have you consider the definition differences and how you deal with this issue? Does it affect the results?

   Response:
   We totally understand the reviewer's concern. The definition of forest is not uniform across these datasets. However, it will not affect the validation samples generation and its quality, because we only sampled from the consensus area (described detailly in Section 3.3). In other words, the undefined areas of forests due to different definitions will not be considered to generate regrowth samples.

2. Line205, 'too large' -> 'too high'. Same to Line206, because sensitivity should be described by high/low.

Response:
We thank the reviewer for pointing out this issue. The statement of 'too large' has been modified accordingly.

*"For example, if the sensitivity is too ==high==, then slow forest degradation (owing to insect pests and selective logging, etc.) will also be detected as breakpoints. Because there is no land cover type change in this process, a ==high== sensitivity will lead to an underestimation of forest age…"*

3. For figure5, why the second column forest grids were all classified as UF? For 2015, forest was identified in the second column of the 2nd and 3rd rows of the both datasets. Seems these two grids are also regrowth forests according to the classifier defined. Also, there is a typo of 'includiing', which shoud be 'including'.

Response:
Thank you so much for your careful check. These two grids you mentioned belong to the regrowth forest, so we modified Figure 5. Also, the spelling mistakes were corrected accordingly.

[Figure]

*Figure 5. Validation samples generated using LULC products.*

4. Line272, I am not very clear how the validation sample sets were generated.

Could you provide more information here?

Response:
Thank you for pointing out this issue. The validation sample were generated by auxiliary datasets (Table 1). First, we confirmed the area of consense regrowth (CR) and consense non-regrowth (CN) with four periods (such as 2000–2005, 2005–2010, 2010–2015 and 2015–2020). Second, regrowth samples and non-regrowth samples were randomly generated from CR and CN of each period, respectively. As a result, we obtained 2618 regrowth samples and 21007 non-regrowth samples.

*"(3) Random sampling and confusion matrix calculation. ==Stratified random sampling was used to generate validation sample sets. First, we confirmed the area of consense regrowth (CR) and consense non-regrowth (CN) with four periods (i.e., 2000–2005, 2005–2010, 2010–2015, and 2015–2020). Second, about 1000 regrowth samples and 5000 non-regrowth samples were randomly generated from CR and CN of each period.== Considering the possibility of regrowth events occurring in each period within the same pixel, only the regrowth samples in the most recent period were retained for the regrowth samples in the four periods. As a result, 2,618 regrowth samples (red dots in Figure 6) and 21,007 non-regrowth samples (blue dots in Figure 6) were obtained."*

5. Line289, 'smaller'->'lower'

Response:
We thank the reviewer for pointing out this issue. The word 'smaller' have been replaced by 'lower'.

*"The provinces with relatively weak classification performance were Gansu, Jiangxi, Shaanxi and Beijing (in order), and the OAs of these four provinces were ==lower== than 60%."*

6. Line308-309, better rephrase this sentence: "more ... than ...". It is not appropriate to compare these two since your data only limited to young forests, while MPI-BGC covers all ages.

Response:
We gratefully appreciate your valuable suggestion. We have rewritten this part according to your suggestion.

*"...depict the age of these forests. ==The forest age map produced in this research presents clear information at the 30 m spatial resolution, which is helpful for monitoring small-scale deforestation activities and estimating land-atmosphere carbon fluxes.=="*

7. Line318, why randomly selected samples but not all the regrowth data?

Response:
Thank you so much for your careful check. We only randomly selected 10000 samples to calculate Pearson's product-moment correlation coefficient, because 10000 samples can basically reflect the relationship between two data. However, using all the regrowth data is time-consuming and we should consider the limits of GEE's computing power.

8. Lines 403-404, This may not be the case. For example, it could be the reason that the forestation areas remained the same but the forest establishment (tree survival rate) was lower in recent decade. To make this claim, you need to refer to the data of forestry yearbook.

Response:
We gratefully appreciate for your valuable comment. Indeed, the lower rate of tree survival after 2000 also could be the reason. So we referred to the $5^{th}$, $6^{th}$, $7^{th}$, and $8^{th}$ national forest inventory data and found that the area of net gain planted forest is 102,520, 65,924, 84,311, and 76,416 $km^2$ during 1994-1998, 1999-2003, 2004-2008, and 2009-2013, respectively. It means that there was less planted forest after 1999, which may be the reason. According to this, we rephrased the sentences of Lines 403-404.

*"We referred to the $5^{th}$, $6^{th}$, $7^{th}$, and $8^{th}$ national forest inventory data and found that the area of net gain planted forest is 102,520, 65,924, 84,311, and 76,416 $km^2$ during 1994-1998, 1999-2003, 2004-2008, and 2009-2013, respectively (Liu et al., 2021). It means that there was less planted forest after 1999, which is consistent with our findings. Another reason may be that the country's early policies (specifically, the Returning Farmland to Forest Program and the Afforestation Program) were implemented effectively, and by 2000 many areas suitable for afforestation had been occupied."*

9. Line426-428, Yes, this is reasonable. Suggest to use eucalyptus, which has been widely planted in Guangxi and Guangdong, as an example. Eucalyptus is a fast growing species and is generally harvested in 5-10 years.

Response:
We thank the reviewer for the nice suggestion. We have re-written this part according to the reviewer's suggestions.

*"On the other hand, a large number of eucalyptus plantations were distributed in southern China, leading to young forest regrowth in the south."*

Response to CC2:

High-resolution forest age mapping is an important part of carbon cycle research and is one of the most significant research points. Based on the CCDC algorithm, this study maps the age of young forests in China with a resolution of 30 meters. This product are valuable for the calculation of the carbon cycle and carbon budget. As a user, I am very interested in your dataset, but I found some limitations in this dataset which may hinder its further applications.

First, it is found that there are serious spatial discontinuity in this dataset, such as the following regions: R1 (121-122°E, 50-51°N), R2 (123-125°E, 51-52°N), R3 (117-119°E, 29-30°N), R4 (119-120°E, 28-29°N).

Second, the forest age mapping was carried out based on the CCDC algorithm, but it has been demonstrated that the CCDC algorithm had several limitations. (1) It did not consider the spatial differences between pixels (Ye et. al., 2023). (2) It did not consider the varied temporal consistency of the Landsat time series (Zhu et. al., 2020). (3) Large inconsistency of disturbance maps existed between the adjacent Landsat path overlap and non-overlap regions (Qiu et. al., 2022 Characterization of land disturbances based on Landsat time series). Why not use a better version of the CCDC-family algorithms such as COLD (Zhu et. al., 2020), Bi-CCD (Zheng et. al., 2021), S-CCD (Ye et. al., 2021), NRT-MONITOR (Shang et. al., 2022), OB-COLD (Ye et. al., 2023)?

Last, it was reported that young forests under 31 years old account for 19% of China's total forest (Lines 400-401, page 16), which is quite different from the results of the ninth forest inventory in China. These differences should be clearly explained.

Response:
We thank you for using our dataset and giving the above positive comments and nice suggestions. We have carefully considered your comments and have responded as follows.

First, this issue is mainly about the set of values for years larger than 31. Specifically, by classifying the pixels with values >31 into one category and then displaying forest age, the problem of spatial discontinuity you mentioned can be resolved. We have resolved this issue and shared a new version of the dataset, which is openly available at https://doi.org/10.6084/m9.figshare.21627023.v7.

*"6 Data availability*
*The produced 30 m map of young forest age across China in this research is openly available at https://doi.org/10.6084/m9.figshare.21627023.v7 (Xiao, 2022). The Landsat data and the auxiliary data are from public data archive and user team of GEE (https://code.earthengine.google.com/)."*

Second, as you mentioned, there are some CCDC-family algorithms, which may be more suitable for turbulence monitoring and/or classification of land cover. We use CCDC to track the breakpoints of Landsats time series, for the three main reasons: (1)

CCDC is the classical algorithm in turbulence detection. We considered using it to estimate the age of forest and the results in this paper already demonstrated that it is an acceptable choice; (2) other CCDC-family algorithms might be sensitive to detect breakpoints. In future research, we will examine whether the use of other versions will necessarily further increase the accuracy; (3) GEE cloud platform provided the basic CCDC in its official algorithm libraries, which is more suitable for large-scale mapping than other CCDC-family algorithms currently.

Third, the differences you mentioned may come from three parts: (1) Differences in statistical time. The ninth national forest inventory (NFI) of China is covering the period 2014–2018, however, our dataset is covering the period 1990–2020; (2) Differences in the methods of forest age statistics. The NFI classified the forest into five forest classes (such as young, mid-aged, near-mature, mature, and over-mature forests), and the age range of each class is vary with tree types. For example, the natural Pinus massoniana Lamb less than 20 years old belongs to the young stage, while the natural Abies fabri less than 40 years old also belongs to the young forest. However, we definite the 1-31-year-old forests as young forests; (3) Mapping error. As mentioned in Section 5.4 of the manuscript, there are still uncertainties in estimating the age of forests.

Response to referee #2:

In this manuscript a continuous change detection and classification (CCDC)-based method for large-scale forest age mapping is proposed, and used to estimate young forest ages across China in 2020 at a spatial resolution of 30 m. This is of interest to the scientific community. The reliability and applicability of the proposed CCDC-based forest age mapping method has been validated by comparing the forest age map with 20 Hansen's forest change dataset, Max Planck Institute for Biogeochemistry (MPI-BGC) 1 km global forest age datasets and field measurements. This study would be very helpful to reduce the uncertainties in the research of forest carbon cycle. It only needs a minor revisions as follows:
1) Line 518: "of should be" should be replaced with "should be".

Response:
Thank you very much for your positive comments and suggestions to improve our manuscript. We have carefully considered your comments and revised the paper accordingly. We have modified the sentence in Line 518.

Thank you again for your work on our paper. We look forward to hearing from you in due course.

With best wishes
The authors

---

## Author Response (AR3)

Dear Editor,

Thank you very much for taking time out of your busy schedule to improve our manuscript. We have carefully considered your comments and revised the paper accordingly. The revised parts are marked in yellow. The replies to the two main issues are as follows:

The audience mentioned the proportion of young forests identified in this work is too low. The authors attributed it to the differences in inventory time and forest age classifications and mapping errors. However, these reasons are not persuasive. At least, the authors can calculate the annual areas of young forests (using varying age thresholds, e.g., 30 years, 40 years, 50 years) and compare them with statistics. In addition, the field validation should use more data (please refer to CPSDv0: A forest stand structure database for plantation forests over China).

Response:
The audience mentioned that the proportion (19%) of young forests identified in our work is quite different from the statistical value (32.67%, Table S1) of the ninth national forest inventory (NFI) of China. We have clarified this from the following three points:

(1) The proportion of 1-31-year-old forests in our product was calculated based on the total forest area (245.20 million hectares) in China, while the proportion of young forests in the 9[th] NFI was calculated based on the total area of arboreal forests (179.89 million hectares, see Table S1) in China. Therefore, if we calculated the proportion of 1-31-year-old forests based on the total area of arboreal forests, the value of our product will be higher (25.69%).

Table S1. Area and standing volume of different age groups of arboreal forests in China (State Forestry Administration of China, 2018)

| Age groups | Area (million ha) | Area ratio (%) | Standing volume (million $m^3$) | Standing volume ratio (%) |
|---|---|---|---|---|
| Young | 58.78 | 32.67 | 2139.14 | 12.54 |
| Mid-aged | 56.26 | 31.27 | 4821.35 | 28.26 |
| Near-mature | 28.61 | 15.91 | 3514.29 | 20.60 |
| Mature | 24.68 | 13.72 | 4011.11 | 23.52 |
| Over-mature | 11.56 | 6.43 | 2572.30 | 15.08 |
| Total | 179.89 | 100.00 | 17058.20 | 100.00 |

(2) The rule of age-group classification in NFI is completely different from our definition of young forest age. According to the regulations formulated by the State Forestry Administration of China on age-class and age-group division of main tree-species, the delineation of different age groups is varied to the tree species, forest

types, origins, and management level (State Forestry Administration of China, 2018). For example, the natural Pinus massoniana from north of China with less than 20 years old belongs to the young stage, while the natural Red Pine from North of China with less than 40 years old also belongs to the young forest (Table S2).

However, we definite the 1-31-year-old forests as young forests, which is different from the definition of the young forest group in NFI. Thus, it is difficult to uniform 1-31-year-old forests in our map with the statistics in NFI due to its limit forest age range (1-31 years) and other forest properties (e.g., tree species, forest types, origin and management level) that are needed in the classification of forest age groups.

Table S2. Age group division of main tree species in general timber forest (State Forestry Administration of China, 2018)

| Tree species | District | Origin | Age groups (unit: years) | | | | |
|---|---|---|---|---|---|---|---|
| | | | Young | Mid-aged | Near-mature | Mature | Over-mature |
| Red Pine, Spruce, Hemlock, Cedar | North | Natural | ≤60 | 61-100 | 101-120 | 121-160 | ≥161 |
| | | Planted | ≤40 | 41-60 | 61-80 | 81-120 | ≥121 |
| | South | Natural | ≤40 | 41-60 | 61-80 | 81-120 | ≥121 |
| | | Planted | ≤30 | 31-50 | 51-60 | 61-80 | ≥81 |
| Cupressus funebris | North | Natural | ≤60 | 61-100 | 101-120 | 121-160 | ≥161 |
| | | Planted | ≤30 | 31-50 | 51-60 | 61-80 | ≥81 |
| | South | Natural | ≤40 | 41-60 | 61-80 | 81-120 | ≥121 |
| | | Planted | ≤30 | 31-50 | 51-60 | 61-80 | ≥81 |
| Larch, Abies fabri, Black Pine, Pinyon Pine | North | Natural | ≤40 | 41-80 | 81-100 | 101-140 | ≥141 |
| | | Planted | ≤20 | 21-30 | 31-40 | 41-60 | ≥61 |
| | South | Natural | ≤40 | 41-60 | 61-80 | 81-120 | ≥121 |
| | | Planted | ≤20 | 21-30 | 31-40 | 41-60 | ≥61 |
| Pinus tabuliformis, Pinus massoniana | North | Natural | ≤30 | 31-50 | 51-60 | 61-80 | ≥81 |
| | | Planted | ≤20 | 21-30 | 31-40 | 41-60 | ≥61 |
| | South | Natural | ≤20 | 21-30 | 31-40 | 41-60 | ≥61 |
| | | Planted | ≤10 | 11-20 | 21-30 | 31-50 | ≥51 |
| Poplar, Willow, Tung tree, Paulownia, Acer negundo | North | Natural | ≤20 | 21-30 | 31-40 | 41-60 | ≥61 |
| | | Planted | ≤10 | 11-15 | 16-20 | 21-30 | ≥31 |
| | South | Natural | - | - | - | - | - |
| | | Planted | ≤5 | 6-.10 | 11-15 | 16-25 | ≥26 |
| Melia azedarach | South | Natural | ≤20 | 21-30 | 31-40 | 41-60 | ≥61 |
| | | Planted | ≤5 | 6-10 | 11-15 | 16-25 | ≥26 |
| Robinia pseudoacacia | North | Regardless of origins | ≤10 | 11-15 | 16-20 | 21-30 | ≥31 |
| | South | | ≤5 | 6-10 | 11-15 | 16-25 | ≥26 |
| Ephedra, | South | Planted | ≤5 | 6-10 | 11-15 | 16-25 | ≥26 |

| Species | Region | Type | | | | | |
|---|---|---|---|---|---|---|---|
| Eucalyptus | | | | | | | |
| Maple Birch, Birch (excluding Black Birch), Elm, Magnolia, Sweetgum | North | Natural | ≤30 | 31-50 | 51-60 | 61-80 | ≥81 |
| | | Planted | ≤20 | 21-30 | 31-40 | 41-60 | ≥61 |
| | South | Natural | ≤20 | 21-40 | 41-50 | 51-70 | ≥71 |
| | | Planted | ≤10 | 11-20 | 21-30 | 31-50 | ≥51 |
| | South | Planted | ≤20 | 21-40 | 41-50 | 51-70 | ≥71 |
| Spruce, Fir, Hemlock | South | Planted | ≤10 | 11-20 | 21-25 | 26-35 | ≥36 |

(3) The forest parameters in the 9[th] NFI were conducted between 2014 and 2018, while the forest area we have calculated is based on the data from 2020. This difference in time period may also result in some discrepancies.

Based on the above analysis, we believe that the proportion of 1-31-year-old forests in the manuscript is reasonable. However, to ensure the rigor of the manuscript and avoid any potential misinterpretation of this value by readers, we have made the following modifications to the relevant description.

*"… To show the spatial distribution of young forest age more clearly, we divided the forest into four stand age classes, namely stand age class I (1–10 years), II (11–20 years), III (21–31 years) and IV (> 31 years). In the 1-31-year-old forests, stand age class III accounted for the largest proportion (39.32%), followed by stand age class II (38.34%). Stand age class I (22.34%) accounted for the smallest proportion."*

In addition, thank you for your constructive suggestions about the field validation. We have carefully read the paper titled "CPSDv0: a forest stand structure database for plantation forests in China" and downloaded its dataset (CPSDv0). It should be noted that we have made pre-processing on this dataset in three aspects:

(1) We updated the forest age in CPSDv0 based on the investigation year of sampling plots. For example, if the sampling time was 2010 and the corresponding recorded forest age was 7 years, then in 2020, the forest age should be 2020-2010+7=17 years. It should be noted that this calculation is based on the assumption that there has been no logging or land use conversion since the survey time of the sampling points.

(2) We filtered out the observation points related to longitude or latitude recorded in decimal degree notation with only two or three decimal places retained because such sampling plots do not include precise geographical coordinates.

(3) Observation points with forest ages older than 31 were also filtered out because we only calculated 1-31-year-old forest in our product.

Then, we used the coordinates of these observation points to find out the predicted

forest age in our product. If the predicted age is less than the value of 2020 minus the year of investigation, we will delete this observation, as we cannot determine whether forest succession has occurred at the observation point after the year of investigation. Finally, we obtained 28 records with accurate geographical locations from CPSDv0. After combining them with the 23 validation points that we previously collected from other studies, we now have a total of 51 field measurements (Table 3). We conducted a new evaluation of forest age based on the updated field measurements. Referring to the field measurements, the predicted forest age has a correlation coefficient of 0.77 and root mean square error (RMSE) of 5.15, suggesting an acceptable correlation with the field measurements (Figure 10). Accordingly, we have updated the relevant descriptions and charts in the manuscript.

*"4.1.3 Evaluation based on field measurements*

*The data of field measurements are composed of two parts. The first part was derived from 150 relevant papers published after 2020 from China National Knowledge Infrastructure (CNKI). We searched them using the following keywords: China and forest age. The second part was derived from Wu et al. (2023). It should be pointed out that three pre-processing steps were performed on this dataset. First, we updated the forest age in field measurements based on the investigation year of sampling plots. For example, if the sampling time was 2010 and the corresponding recorded forest age was 7 years, then in 2020, the forest age should be 2020-2010+7=17 years. It should be noted that this calculation is based on the assumption that there has been no logging or land use conversion since the survey time of the sampling points. Second, we filtered out the observation points related to longitude or latitude recorded in decimal degree notation with only two or three decimal places retained, because no precise geographical coordinates are available for these sampling plots without. Third, observation points with forest ages older than 31 were also filtered out because we only calculated 1-31-year-old forest in our product.*

*Then, we used the coordinates of these observation points to find out the predicted forest age in our product. If the predicted age is less than the value of 2020 minus the year of investigation, we will delete this observation, as we cannot determine whether forest succession has occurred at the observation point after the year of investigation. Finally, we obtained 51 field measurements (Table 3) with accurate geographical locations. Figure shows the scatter plot between the field measurements and predicted forest age. Referring to the field measurements, the predicted forest age has a correlation coefficient of 0.77 and root mean square error (RMSE) of 5.15, suggesting an acceptable correlation with the field measurements."*

[Figure]

**Figure 10.** Comparison between the forest age derived from field measurements (observed forest age) and predicted forest age.

Table 3. Information on the 51 field measurements.

| ID | Longitude | Latitude | Observed forest age | Predicted forest age | Year of investigation | Source |
|---|---|---|---|---|---|---|
| 1 | 109.328858 | 23.050233 | 3 | 3 | 2021 | Li et al. (2021) |
| 2 | 109.332939 | 23.053525 | 8 | 8 | 2021 | Li et al. (2021) |
| 3 | 109.242036 | 23.111756 | 18 | 16 | 2021 | Li et al. (2021) |
| 4 | 109.160242 | 23.053275 | 21 | 25 | 2021 | Li et al. (2021) |
| 5 | 109.159194 | 23.040914 | 29 | 34 | 2021 | Li et al. (2021) |
| 6 | 122.491287 | 42.717326 | 20 | 9 | 2015 | Han et al. (2022) |
| 7 | 122.571380 | 42.684847 | 30 | 35 | 2015 | Han et al. (2022) |
| 8 | 113.421000 | 23.245000 | 6 | 6 | 2020 | Chen et al. (2022) |
| 9 | 113.393000 | 23.226000 | 10 | 23 | 2020 | Chen et al. (2022) |
| 10 | 113.419000 | 23.256000 | 15 | 18 | 2020 | Chen et al. (2022) |
| 11 | 113.394000 | 23.212000 | 20 | 13 | 2020 | Chen et al. (2022) |
| 12 | 113.381000 | 23.255000 | 30 | 27 | 2020 | Chen et al. (2022) |
| 13 | 106.740000 | 26.520000 | 11 | 12 | 2019 | Yin et al. (2021) |
| 14 | 110.465833 | 22.048333 | 5 | 5 | 2020 | Song et al. (2021) |
| 15 | 110.500833 | 21.919167 | 15 | 15 | 2020 | Song et al. (2021) |
| 16 | 110.500278 | 22.022222 | 5 | 7 | 2020 | Song et al. (2021) |
| 17 | 110.517500 | 21.908056 | 15 | 8 | 2020 | Song et al. (2021) |
| 18 | 110.516111 | 21.908056 | 10 | 1 | 2020 | Song et al. (2021) |
| 19 | 117.935278 | 26.881389 | 7 | 9 | 2017 | Feng et al. (2021) |
| 20 | 118.451667 | 26.243333 | 2 | 7 | 2020 | Hong et al. (2021) |
| 21 | 116.650833 | 25.172778 | 3 | 9 | 2020 | Hong et al. (2021) |

| | | | | | | |
|---|---|---|---|---|---|---|
| 22 | 118.351389 | 27.317500 | 7 | 12 | 2020 | Hong et al. (2021) |
| 23 | 117.802222 | 27.275556 | 9 | 17 | 2020 | Hong et al. (2021) |
| 24 | 104.5672222 | 28.60166667 | 17 | 15 | 2011 | Wu et al. (2023) |
| 25 | 104.5769 | 28.6093 | 8 | 5 | 2015 | Wu et al. (2023) |
| 26 | 106.8760472 | 22.06267778 | 13 | 11 | 2013 | Wu et al. (2023) |
| 27 | 106.9072889 | 22.02632778 | 23 | 15 | 2013 | Wu et al. (2023) |
| 28 | 106.910175 | 22.02430833 | 23 | 17 | 2013 | Wu et al. (2023) |
| 29 | 106.9112 | 22.03783056 | 13 | 13 | 2013 | Wu et al. (2023) |
| 30 | 106.9132222 | 22.02641667 | 23 | 23 | 2013 | Wu et al. (2023) |
| 31 | 108.1666667 | 22.86666667 | 17 | 15 | 2012 | Wu et al. (2023) |
| 32 | 109.1713889 | 36.07972222 | 30 | 19 | 2015 | Wu et al. (2023) |
| 33 | 109.2833333 | 21.96666667 | 22 | 20 | 2012 | Wu et al. (2023) |
| 34 | 109.3582222 | 19.51252778 | 13 | 16 | 2012 | Wu et al. (2023) |
| 35 | 109.4833333 | 23.91666667 | 17 | 19 | 2009 | Wu et al. (2023) |
| 36 | 109.6075556 | 26.69930556 | 13 | 15 | 2010 | Wu et al. (2023) |
| 37 | 109.6076667 | 26.70025 | 13 | 13 | 2010 | Wu et al. (2023) |
| 38 | 109.8933333 | 24.76333333 | 13 | 7 | 2012 | Wu et al. (2023) |
| 39 | 110.1018333 | 21.26166667 | 6 | 13 | 2015 | Wu et al. (2023) |
| 40 | 110.10185 | 21.26188333 | 7 | 13 | 2015 | Wu et al. (2023) |
| 41 | 110.4028833 | 34.0909 | 17 | 13 | 2012 | Wu et al. (2023) |
| 42 | 110.6969444 | 30.91891667 | 25 | 15 | 2015 | Wu et al. (2023) |
| 43 | 112.8481306 | 27.29384722 | 11 | 12 | 2013 | Wu et al. (2023) |
| 44 | 112.8485611 | 27.29428611 | 10 | 16 | 2013 | Wu et al. (2023) |
| 45 | 113.3548833 | 27.35978889 | 11 | 12 | 2013 | Wu et al. (2023) |
| 46 | 113.3865194 | 27.35451667 | 18 | 10 | 2013 | Wu et al. (2023) |
| 47 | 116.4591167 | 25.63750278 | 17 | 15 | 2011 | Wu et al. (2023) |
| 48 | 117.5247222 | 26.81388889 | 21 | 17 | 2014 | Wu et al. (2023) |
| 49 | 117.5408333 | 26.80722222 | 16 | 14 | 2014 | Wu et al. (2023) |
| 50 | 119.8430556 | 30.24833333 | 31 | 29 | 2014 | Wu et al. (2023) |
| 51 | 122.5455556 | 52.97833333 | 26 | 29 | 2010 | Wu et al. (2023) |

The correction of spatial discontinuity should give clear figures and clarify why there were sharp edges between forests of older than 30 years and those of younger than 30 years.

Response:
Thank you very much for your comments and suggestions. The reason for the spatial discontinuity is that in our previous version, we did not unify the pixels greater than 31 years into one category. That is, we did not mask the areas with forest ages over 31 years, resulting in spatial discontinuity of the product. The reason for the existence of >31 years forest is that in some areas, data from 1985 are available. Thus, for these areas, we can estimate forest age of 32-35 years. However, some areas in China do

not have images before 1990, so only young forests under 31 years old can be mapped in these areas.

In the new version, to ensure the consistency of the forest age range nationwide, the forest age range we produced has been set to 1-31 years. That is, for all areas with ages larger than 31 years, we just set a uniform value presenting the meaning of $> 31$ years. This problem has been solved in the new version of the product, which is now openly available at https://doi.org/10.6084/m9.figshare.21627023.v7. Figures S2-S5 show the initial version of the dataset of forest age and its new version in four regions.

[Figure]

Figure S2. Initial version (a) of the dataset of forest age and its new version (b) in region 1 (R1).

[Figure]

Figure S3. Initial version (a) of the dataset of forest age and its new version (b) in region 2 (R2).

[Figure]

Figure S4. Initial version (a) of the dataset of forest age and its new version (b) in region 3 (R3).

[Figure]

Figure S5. Initial version (a) of the dataset of forest age and its new version (b) in region 4 (R4).

Thank you again for your work on our paper. We look forward to hearing from you in due course.

With best wishes
The authors

**References**
State Forestry Administration of China. *"Regulations for age-class and age-group division of main tree-species."* (2018).